# LESS GRADIENT, MORE SPEED: RETHINKING PIPELINE PARALLELISM FOR EFFICIENT FINE-TUNING WITH FLUIDPIPE

## ABSTRACT

Fine-tuning large pretrained models often uses pipeline parallelism (PP) to split layers across devices. PP is simple to deploy but requires per-iteration cross-stage gradient exchanges, creating pipeline bubbles that reduce efficiency and making performance highly sensitive to latency. We introduce **FluidPipe**, a two-stage pipeline design that replaces these gradient exchanges with local updates guided by an auxiliary head and cross-stage bi-directional distillation. This re-design eliminates iteration-time synchronization while preserving model quality. We develop a cost and communication model explaining when FluidPipe outperforms PP, and validate on BERT-Large and ViT-Large fine-tuning, where FluidPipe achieves up to $3.3\times$ faster training while matching or improving accuracy.

## 1 INTRODUCTION

Fine-tuning modern foundation models entails support parallelized execution of computational graphs for models that can span billions of parameters— examples include GPT-3 (175B parameters) (Brown et al., 2020), PaLM (540B) (Chowdhery et al., 2023), LLaMA-2 (7B-70B) (Touvron et al., 2023), and GLaM (1.2T, mixture-of-experts) (Du et al., 2022) on the language side, as well as vision backbones such as Swin Transformer V2 (3B parameters) (Liu et al., 2022) and large ViT variants (Dosovitskiy et al., 2021). Fine-tuning does not reduce the capacity requirements of the underlying model: the full parameter set must still be stored and trained.[1] As a result, a single accelerator is often insufficient, and practitioners must rely on multiple GPUs or nodes to execute fine-tuning efficiently.

Practitioners address these requirements using combinations of data parallelism, tensor/model parallelism, optimizer sharding, and pipeline parallelism (PP) (Shoeybi et al., 2019; Rajbhandari et al., 2020). Among these, PP is especially attractive for fine-tuning because it is easy to deploy: the model is partitioned into sequential stages mapped across devices, and mini-batches are split into micro-batches to overlap forward and backward passes. However, each micro-batch still incurs two synchronizations per stage boundary—activations in the forward pass and gradients in the backward pass. These fine-grained communications create *pipeline bubbles*, idle gaps where stages wait for transfers to complete. Even within a datacenter, bubbles reduce throughput, and in cross-node or cross-region settings (which commonly arise due to poor availability of co-located resources) they dominate runtime (Strati et al., 2024). Such inefficiency directly inflates wall-clock time and compute cost.

Prior work has focused on *scheduling around bubbles*. GPipe (Huang et al., 2019) and Megatron (Shoeybi et al., 2019) overlap work using micro-batches, PipeDream (Narayanan et al., 2019) and PipeMare (Yang et al., 2021) employ asynchronous updates, and Zero-Bubble PP (Qi et al., 2023) and BitPipe (Wu et al., 2024) refine micro-batch interleaving. These approaches reduce stalling but cannot escape the *fundamental requirement* that every iteration must return gradients across stage boundaries. As a result, they remain sensitive to latency and bandwidth. This observation raises a natural question: can pipeline parallelism be redesigned to *avoid per-iteration gradient dependencies across stages* without sacrificing accuracy?

---

[1]We focus on full-model fine-tuning; parameter-efficient fine-tuning (PEFT) methods such as adapters or LoRA (Houlsby et al., 2019; Hu et al., 2022; Pfeiffer et al., 2020) can reduce memory and compute needs, but are not always applicable or optimal. We include LoRA as a baseline in our experiments.

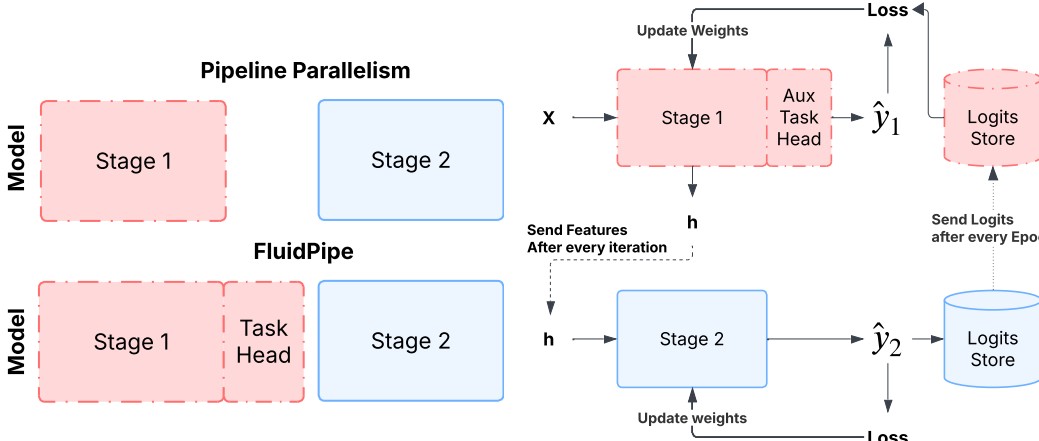

(a) Structural comparison. Both PP and FluidPipe use a two-stage split; FluidPipe additionally places a small auxiliary head on Stage 1, enabling local training at it.

(b) FluidPipe overview. At each iteration, Stage 1 computes features $h = S_1(x)$, sends $h$ and auxiliary logits $\hat{y}_1 = g(h)$ to Stage 2, and updates locally. Stage 2 computes $\hat{y}_2 = S_2(h)$, updates locally, and accumulates logits $\ell_2(x)$ that are sent back *once per epoch* for Stage 1's distillation in the next epoch. No per-iteration gradients cross the stage boundary.

Figure 1: FluidPipe augments a standard two-stage pipeline with an auxiliary head at Stage 1 that removes the need for per-iteration cross-stage gradients.

To this end, we introduce **FluidPipe (FP)**, a pipeline-style training algorithm that removes per-iteration gradient transfers. FP augments the first stage of a standard two-stage pipeline with an auxiliary task head (cf. Figure 1a) so that both stages can update model parameters locally. Cross-stage feedback is provided at low frequency via *bi-directional distillation*: Stage 1 sends auxiliary logits each iteration, while Stage 2 bulk-sends its logits once per epoch. Figure 1b shows an overview of the algorithm. Thus, iteration-time training is fully local within each stage, while feedback is coarse and low-frequency. This design eliminates iteration-time gradient synchronization and reduces sensitivity to round-trip-time (RTT) between stages. Furthermore, FluidPipe demonstrates that rethinking the pipeline dependencies—rather than only optimizing schedules—opens a new path for optimizing pipeline parallelism algorithms.

We restrict our study to the two-stage case in order to isolate the core dependency change—removing per-iteration cross-stage gradients—and to reflect common inter-node fine-tuning deployments. Extending FluidPipe to deeper pipelines is nontrivial: every intermediate stage would need to train under its own auxiliary head, and it remains an open question how much representational capacity such stages retain and how their local learning interacts with the global task. In addition, one must design a distillation protocol (hierarchical vs. pairwise) and synchronization policy that preserves both accuracy and efficiency. We leave these algorithmic questions for future work. Nonetheless, Section 5.1 illustrates FP's compatibility with pipelines beyond two stages by combining intra-node PP with inter-node FP in a mixed topology.

In summary, our contributions are:

- We propose FluidPipe, a two-stage pipeline design that eliminates per-iteration gradient transfers via bi-directional distillation (Section 3).

- We provide a cost model and communication analysis showing when FP outperforms PP (Section 4).

- We empirically validate FP on ViT-Large and BERT-Large fine-tuning across datacenter and cross-region latencies, achieving 1.5–2.4× speedups while preserving accuracy (Section 5).

---

**Algorithm 1** FluidPipe: Stage 1 (Partial Model) Procedure

---

Initialize $\theta_1^o$, $\mathcal{P}_2 \leftarrow \{\}$ $\qquad\qquad\qquad\qquad$ ▷ $\mathcal{P}_2$ denotes Stage 2 logits keyed by sample index
**for** epoch $\leftarrow 1$ **to** $E$ **do**
$\quad$ **for each** mini-batch $(x, y, i)$ from dataset $\mathcal{D}$ **do** $\qquad\qquad$ ▷ $i$ denotes sample indices
$\qquad$ **1.** $h \leftarrow S_1(x; \theta_1^h)$ $\qquad\qquad\qquad\qquad\qquad\qquad$ ▷ Intermediate features
$\qquad$ **2.** $\hat{y}_1 \leftarrow g(h; \theta_1^o)$ $\qquad\qquad\qquad\qquad$ ▷ Auxiliary head (Stage 1 logits)
$\qquad$ **3. Send** $(h, \hat{y}_1, i)$ to Stage 2 $\qquad\qquad\qquad\qquad\qquad$ ▷ Non-blocking
$\qquad$ **4.** $\mathcal{L}_{\text{total}} \leftarrow \mathcal{L}_{\text{task}}(y, \hat{y}_1)$
$\qquad$ **if** epoch $> 1$ **then**
$\qquad\qquad$ **5.** $\hat{y}_2 \leftarrow \mathcal{P}_2[i]$ $\qquad\qquad\qquad\qquad$ ▷ Stage 2 logits from prior epoch
$\qquad\qquad$ **6.** $\mathcal{L}_{\text{total}} \leftarrow \mathcal{L}_{\text{total}} + \mathcal{L}_{\text{KD}}(\hat{y}_1, \hat{y}_2)$
$\qquad$ **7.** `Backward` and update $\theta_1 \leftarrow \theta_1 - \eta \, \nabla_{\theta_1} \mathcal{L}_{\text{total}}$

$\quad$ **8. Receive** $\mathcal{P}_2$ from Stage 2 $\qquad\qquad\qquad\qquad$ ▷ One blocking receive per epoch
**Output:** parameters $\theta_1^h$

---

## 2 RELATED WORK

**Pipeline Parallelism.** The dominant research line in pipeline parallelism has sought to optimize *scheduling*. GPipe (Huang et al., 2019) introduced micro-batching to overlap forward/backward passes. PipeDream (Narayanan et al., 2019) and PipeMare (Yang et al., 2021) relaxed synchronization via asynchronous schedules. Zero-Bubble PP (Qi et al., 2023) and BitPipe (Wu et al., 2024) refine micro-batch interleaving to shrink idle bubbles. All of these methods retain the fundamental gradient dependency across stages.

In contrast, FluidPipe is *orthogonal*: it eliminates the need for per-iteration gradient exchanges altogether. Scheduling optimizations could be applied within each FluidPipe stage, but FluidPipe's contribution lies in *rethinking the dependency*, not the schedule. This distinction explains why we focus our experiments on comparing FluidPipe to standard PP, while positioning it as complementary rather than competing with advanced scheduling.

**Optimizations to Pipeline Parallelism.** Subsequent work has sought to reduce the impact of pipeline bubbles through more sophisticated scheduling. Zero-Bubble PP (Qi et al., 2023) rearranges the order of forward and backward micro-batches to remove idle gaps, while BitPipe (Wu et al., 2024) proposes bidirectional and interleaved schedules to increase overlap between stages. These methods focus on carefully orchestrating computation to minimize bubbles, but they do not alter the fundamental synchronization requirement that gradients must be exchanged across stages at every iteration.

**Other fine-tuning strategies.** An alternative to multi-GPU training is using parameter-efficient fine-tuning (PEFT). LoRA (Hu et al., 2022) and related methods (e.g., adapters (Houlsby et al., 2019; Pfeiffer et al., 2020)) update only a small subset of parameters, allowing models to be fine-tuned on a single GPU. We include LoRA as a baseline in our experiments to contrast communication-efficient distributed training (FluidPipe) with compute-efficient local fine-tuning.

## 3 FLUIDPIPE DESIGN

### 3.1 OVERVIEW

Pipeline parallelism (PP) splits a model into stages and processes a mini-batch as $m$ micro-batches. In each iteration, every cross-stage boundary incurs an *activation send* in the forward pass and a *gradient return* in the backward pass for each micro-batch. These exchanges couple the stages in both directions: a stage cannot start the backward pass for micro-batch $i$ until the downstream stage finishes its forward on $i$ and returns the gradient. Figure 2 illustrates an iteration with two stages and two micro-batches.

---

**Algorithm 2** FluidPipe: Stage 2 (Full Model) Procedure

---

> **for** epoch $\leftarrow 1$ **to** $E$ **do**
>> Initialize $\mathcal{P}_2 \leftarrow \{\}$        $\triangleright$ Accumulate Stage 2 logits for epoch-level send
>> **for each** mini-batch $(x, y, i)$ **do**        $\triangleright$ Paired with Stage 1 stream
>>> **1. Receive** $(h, \hat{y}_1, i)$ from Stage 1
>>> **2.** $\hat{y}_2 \leftarrow S_2(h; \theta_2)$        $\triangleright$ Full-model logits
>>> **3.** $\mathcal{L}_{\text{total}} \leftarrow \mathcal{L}_{\text{task}}(y, \hat{y}_2) + \mathcal{L}_{\text{KD}}(\hat{y}_2, \hat{y}_1)$  $\triangleright$ Distill from Stage 1 into Stage 2 each iteration
>>> **4.** `Backward` and update $\theta_2 \leftarrow \theta_2 - \eta \nabla_{\theta_2} \mathcal{L}_{\text{total}}$
>>> **5.** $\mathcal{P}_2[i] \leftarrow \hat{y}_2$        $\triangleright$ Store logits for epoch-level send
>>
>> **6. Send** $\mathcal{P}_2$ to Stage 1        $\triangleright$ One blocking send per epoch
> **Output:** parameters $\theta_2$

---

What makes these fine-grained synchronizations costly is that with $P$ stages a micro-batch crosses $(P-1)$ boundaries twice (forward and backward), so the latency term on the critical path grows roughly with $2(P-1) \times \text{RTT}$. Increasing $m$ reduces the relative cost of warm-up, but it does not eliminate the per-boundary round-trips that gate backward progress and the optimizer step. Consequently, as the number of boundaries or the RTT increases, these per-iteration synchronizations induce larger pipeline bubbles and lower throughput.

**FluidPipe (FP)** removes the per-iteration gradient dependency. We partition the model into *two* stages. Stage 1 produces features $h = S_1(x)$ and, via an auxiliary head $g(\cdot)$, logits $\hat{y}_1 = g(h)$; it updates from a local loss without cross-stage gradients. Stage 2 receives $(h, \hat{y}_1)$, computes $\hat{y}_2 = S_2(h)$, and updates from its own local loss. Thus, creating a single one-directional exchange per iteration (batch).

To compensate for the missing cross-stage gradients, FP uses *bidirectional distillation*: (i) Stage 2 sends its logits $\ell_2(x)$ to Stage 1 **once per epoch** (teacher-for-Stage 1), and (ii) Stage 1 sends $\hat{y}_1$ **each iteration** alongside $h$ (teacher-for-Stage 2). This keeps iteration-time training fully local on both stages and shifts cross-stage feedback to a coarser, once-per-epoch synchronization. Figure 3 shows how FP communicates and synchronizes between the two stages. Figure 12 shows a timeline of the computations of FP vs PP.

### 3.2 TRAINING MECHANICS

At iteration granularity, Stage 1 minimizes

$$\mathcal{L}_1 = \alpha_1 \mathcal{L}_{\text{task}}(y, \hat{y}_1) + (1 - \alpha_1) \mathcal{L}_{\text{KD}}(\hat{y}_1, \ell_2(x)),$$

where $\ell_2(x)$ are Stage 2 logits received at the end of the previous epoch. For epoch $e{=}1$, we disable distillation by setting $\alpha_1 = 1$, since Stage 1 only receives logits after the end of the first epoch.

Stage 2 minimizes, for each received $(h, \hat{y}_1)$,

$$\mathcal{L}_2 = \alpha_2 \mathcal{L}_{\text{task}}(y, \hat{y}_2) + (1 - \alpha_2) \mathcal{L}_{\text{KD}}(\hat{y}_2, \hat{y}_1),$$

and accumulates $\hat{y}_2$ in a dictionary $\mathcal{P}_2$ keyed by sample index $i$. Once per epoch, Stage 2 *bulk-sends* $\mathcal{P}_2$ to Stage 1, which then uses $\ell_2(x)$ in the next epoch's $\mathcal{L}_1$. Algorithms 1 and 2 provide the exact procedures.

**Design Enhancement: Extra Block**    Stage 1 outputs features $h$ that are both forwarded to Stage 2 and used by the auxiliary head. These two roles can pull the features in different directions. We add an *extra backbone block* after the last Stage 1 block: $h$ is forwarded to Stage 2 unchanged, while $\tilde{h} = f_{\text{extra}}(h)$ is fed to the auxiliary head. This decouples features for continuation from features for local classification. See Figure 7 for its effect across split points.

### 3.3 EXTENSIONS

**Communication policy: bulk vs. streaming.**    We use epoch-level (bulk) transfers of Stage 2 logits to Stage 1 for simplicity and low control overhead. A straightforward extension is to *stream* or

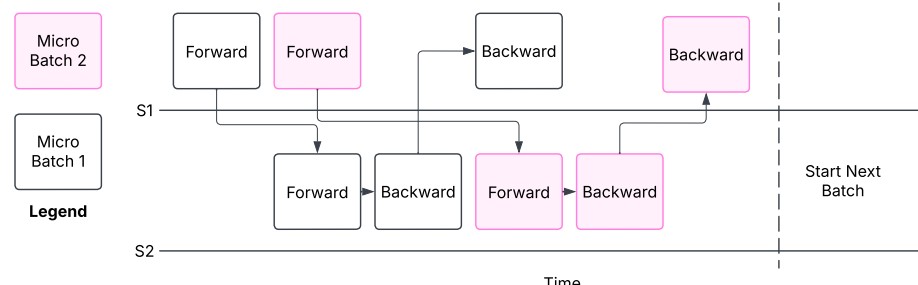

Figure 2: Timeline diagram of Pipeline Parallelism showing cross-stage communication and synchronization with two micro-batches and two stages.

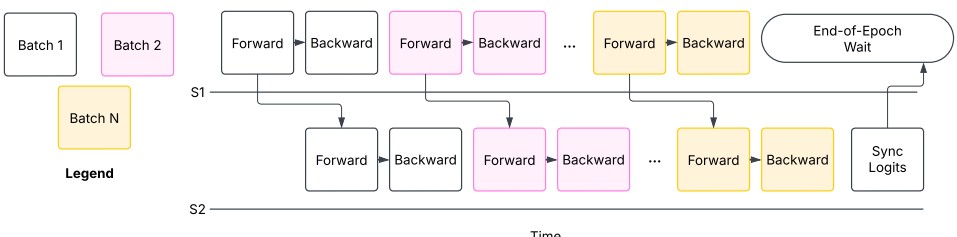

Figure 3: Timeline diagram of FluidPipe showing the cross-stage communication and synchronization over an epoch.

*periodically* send logits during an epoch. Streaming the logits would eliminate the send time at the end of the epoch as well as reduce the memory cost for storing the epochs at Stage 2. Also, it would allow Stage 1 to start the next epoch earlier, since it received logits from Stage 2. Since progress is gated by when Stage 2 finishes the epoch, earlier (streamed) logits to Stage 1 do not shorten the critical path; any speedup over a bulk transfer is likely marginal. We leave the systematic exploration of the different communication policies and their system performance to future work.

**FP without distillation.** When distillation is disabled on a stage, the corresponding messages can be omitted: (i) if Stage 1 uses no distillation (i.e., $\alpha_1=1$), the end-of-epoch Stage 2→Stage 1 logits transfer is unnecessary; (ii) if Stage 2 uses no distillation ($\alpha_2=1$), Stage 1 need only send features $h$ (no $\hat{y}_1$) each iteration. Our current implementation always includes the logits for simplicity, so the reported runtimes for "FP without distillation" are *conservative (upper bounds)*; removing these data would further reduce communication volume without affecting the model quality.

## 4 ANALYTICAL COST MODEL

We analyze the cost of FluidPipe against traditional two-stage pipeline parallelism. Pipeline parallelism divides computation into micro-batches ($m$) to overlap communication and computation. Denote forward-backward compute times at stages 1 and 2 as $t_1$ and $t_2$, respectively. Let $t_a$ and $t_g$ represent communication time for activation and gradient transmissions, respectively. For one epoch of $N_b$ mini-batches, pipeline parallelism takes:

$$T_{\text{PP}} \approx N_b(m+1)\max(t_1 + t_a,\ t_2 + t_g).$$

FluidPipe eliminates per-micro-batch gradient synchronization by performing a one-way data transfer and uses a single epoch-end bulk synchronization. Let $\tau_1, \tau_2$ denote the per-batch computation time in FluidPipe's stage 1 and 2, respectively. Note this includes the time for knowledge distillation. Let $\tau_a$ be the forward transfer time per mini-batch, and $\tau_d$ the epoch-end logits transfer time.

Then, FluidPipe's per-epoch time (accounting for concurrency) is:

$$T_{\text{FP}} \approx N_b \max(\tau_1,\ \tau_2 + \tau_a) + \tau_d.$$

The advantage of FluidPipe over pipeline parallelism primarily comes from removing the frequent per-micro batch gradient synchronization ($t_g$), replacing it with a single bulk synchronization per epoch. Typically, we have $\tau_d \ll N_b m t_g$, leading FluidPipe to outperform pipeline parallelism significantly.

Empirically, the model tracks epoch runtime closely when instantiated with trace-derived step times. In FluidPipe, iteration-time progress is governed by the stage on the critical path (stage 2), so a simple estimate for training time per epoch is

$$\widehat{T}_{\text{epoch}}^{\text{FP}} \approx N_b \left( \tau_2 + \tau_a \right) \quad \text{(optionally } + \ \tau_d \text{ if not negligible).}$$

We obtained $(\tau_2 + \tau_a)$ by measuring the per-step delta on Stage 2 from the instrumented traces and multiplying by the number of batches $N_b$. The resulting $\widehat{T}_{\text{epoch}}^{\text{FP}}$ closely matched the mean epoch runtime reported in our results (see Tables 1 and 4), corroborating that once $\tau_1, \tau_2$ (and the small $\tau_d$) are measured, the cost model accurately predicts training time.

### 4.1 Communication Overhead Analysis

Now we compare the communication volume of FluidPipe and Pipeline Parallelism. Let $\lambda_{\text{batch}}$ denote the per–mini-batch cost of sending the intermediate features $h$ (incurred by both FluidPipe and classical PP), $\lambda_p$ the per–mini-batch cost of sending the auxiliary logits $\hat{y}1$ (FluidPipe only), $\beta\text{batch}$ the per–mini-batch cost of returning backward gradients across a stage boundary (PP only), $\gamma$ the end-of-epoch bulk transfer of Stage 2 logits in FluidPipe, and $N_b$ the number of mini-batches per epoch; for simplicity, we ignore micro-batching.

**Two-Stage Pipeline Parallelism:**
$$\text{Total communication per epoch} \ = \ N_b \times (\lambda_{\text{batch}} + \beta_{\text{batch}}).$$

**FluidPipe:**
$$\text{Total communication per epoch} \ = \ N_b \times (\lambda_{\text{batch}} + \lambda_p) \ + \ \gamma.$$

For FluidPipe to incur lower communication, we need:

$$N_b \left( \lambda_{\text{batch}} + \lambda_p \right) + \gamma \ < \ N_b \left( \lambda_{\text{batch}} + \beta_{\text{batch}} \right)$$
$$\iff \quad (N_b \, \lambda_p) + \gamma < N_b \, \beta_{\text{batch}}.$$

Note that volume of $\gamma$ is the same as $N_b \, \lambda_p$, so we can simplify further and say FluidPipe will incur lower communication if and only if:

$$2(N_b \, \lambda_p) < N_b \, \beta_{\text{batch}}$$

Since the gradient tensor size of a model (e.g., BERT has gradient tensor of size $(b, 512, 1024)$, while the logits tensor size would be $(N_b \times b \times \text{number of labels})$) often far exceeds the size of the logits, thus $2(N_b \, \lambda_p) < N_b \, \beta_{\text{batch}}$ almost always holds in practice.

However, in tasks with very large output spaces—e.g., autoregressive language modeling where the label space equals the vocabulary (often $10^4$–$10^5$ tokens)—the per-sample logit vector can dominate the communication budget, potentially violating the inequality above. In such regimes, *FluidPipe can operate without cross-stage distillation* by setting $\alpha_1 = \alpha_2 = 1$, which removes both the per-iteration transfer of $\hat{y}_1$ and the epoch-end transfer of $\ell_2(x)$. Empirically, our **FP-ND** variant (no distillation) *matches or nearly matches* the best FP configurations in accuracy across vision and language tasks(see Tables 2 and 5). Thus, distillation is an *optional enhancement* rather than a requirement: when logits would be expensive (e.g., large-vocab LM), practitioners can disable it and still obtain FluidPipe's core benefit—eliminating per-iteration gradient synchronization.

## 5 Experiments & Results

**Goals and Setup.** We ask two questions: *(Q1) Does FluidPipe reduce epoch time versus pipeline parallelism (PP) across latencies?* and *(Q2) Does FluidPipe preserve model quality?* We evaluate two settings: *Setup A* fine-tunes ViT-Large/16 (Dosovitskiy et al., 2021) on two machines, each with two A100s (intra-node 2-way PP; inter-node FP, 4 stages in total); *Setup B* fine-tunes BERT-Large (Devlin

Table 1: Mean per-epoch runtime: shown across datasets and latencies (less is better). Speedup over PP is shown in parentheses.

| Task | CIFAR-100 (minutes) | | | Oxford Flowers-102 (seconds) | | | Oxford-IIIT Pets (seconds) | | |
|---|---|---|---|---|---|---|---|---|---|
| Latency | 0.01 ms | 25 ms | 50 ms | 0.01 ms | 25 ms | 50 ms | 0.01 ms | 25 ms | 50 ms |
| PP | 6.04 | 21.88 | 29.99 | 0.13 | 0.46 | 0.69 | 1.86 | 6.67 | 10.05 |
| FP-DB | 3.77 (×1.6) | 9.96 (×2.2) | 14.58 (×2.06) | **0.1** (×1.35) | **0.23** (×1.96) | **0.37** (×1.86) | **1.21** (×1.54) | **2.73** (×2.45) | **4.45** (×2.26) |
| FP-DT | **3.76** (×1.61) | **9.94** (×2.2) | **14.55** (×2.06) | 0.1 (×1.33) | 0.24 (×1.93) | 0.38 (×1.83) | **1.21** (×1.54) | 2.74 (×2.43) | 4.47 (×2.25) |
| FP-DT-EL | 3.77 (×1.6) | 9.96 (×2.2) | 14.58 (×2.06) | 0.1 (×1.31) | 0.24 (×1.9) | 0.38 (×1.8) | 1.23 (×1.52) | 2.77 (×2.41) | 4.52 (×2.22) |
| FP-ND | 3.77 (×1.6) | **9.94** (×2.2) | 14.56 (×2.06) | 0.1 (×1.35) | **0.23** (×1.96) | 0.37 (×1.85) | 1.22 (×1.53) | 2.75 (×2.43) | 4.48 (×2.24) |
| FP-ND-EL | 3.77 (×1.6) | 9.96 (×2.2) | 14.58 (×2.06) | 0.1 (×1.32) | 0.24 (×1.91) | 0.38 (×1.81) | 1.22 (×1.52) | 2.76 (×2.41) | 4.51 (×2.23) |

Table 2: Best accuracy (@epoch) across datasets. Mean with standard deviation is reported.

| | CIFAR-100 | Oxford Flowers-102 | Oxford-IIIT Pets |
|---|---|---|---|
| FP-DB | $93.25 \pm 0.18\%$ | **99.29** $\pm 0.10\%$ | $93.55 \pm 0.06\%$ |
| FP-DT | $93.30 \pm 0.08\%$ | $99.26 \pm 0.02\%$ | **93.87** $\pm 0.21\%$ |
| FP-DT-EL | **93.54** $\pm 0.08\%$ | **99.29** $\pm 0.03\%$ | $93.61 \pm 0.22\%$ |
| FP-ND | $93.21 \pm 0.11\%$ | $99.21 \pm 0.05\%$ | $93.44 \pm 0.37\%$ |
| FP-ND-EL | $93.41 \pm 0.14\%$ | $99.25 \pm 0.13\%$ | $93.48 \pm 0.22\%$ |
| PP | $93.37 \pm 0.24\%$ | $99.11 \pm 0.07\%$ | $93.47 \pm 0.16\%$ |

et al., 2019) on two machines, one A100 each (inter-node FP). We emulate 0.01 ms (same-rack), 25 ms (cross-zone), and for Setup A also 50 ms (inter-region) RTT using Linux tc. All runs use three seeds.

Our design space toggles: (i) distillation on/off and weight $(\alpha_1, \alpha_2)$, and (ii) the *extra block* (Section 3.2). We evaluate five FP variants that separate the effects of distillation from the extra block: **FP-DB** ($\alpha_1 = \alpha_2 = 0.5$), **FP-DT** ($\alpha_1 = \alpha_2 = 0.9$ from Figure 6), **FP-DT-EL** (DT+extra block), **FP-ND** (no distillation), **FP-ND-EL** (ND+extra block). This set yields simple recipes (e.g., ND for zero tuning; DT for light tuning; DT-EL when adding the extra block).

## 5.1 SETUP A: VIT-LARGE FINE-TUNING ON FOUR GPUS

We fine-tune ViT-Large/16 on CIFAR-100, Oxford-IIIT Pets, and Flowers-102 following the original ViT hyperparameters (Dosovitskiy et al., 2021) and for the same number of epochs reported by the ViT paper or until early stopping is triggered. We evaluate test accuracy every 2 epochs on CIFAR-100, every 28 epochs on Oxford-IIIT Pets, and every 100 epochs on Flowers-102. More frequent evaluation would be prohibitively expensive for Oxford-IIIT Pets and Flowers-102, where epochs are very short under large batch sizes. We compare pipeline parallelism and the FluidPipe variants. Within each FluidPipe stage, we use the intra-node PP schedule used by the PP baseline, while employing FP for the inter-node parallelism.

**Epoch-Time Results.** Table 1 reports mean per-epoch time. FluidPipe consistently shortens epochs versus PP at all RTTs. At 0.01 ms, FP variants deliver $\sim 1.49\times$ average speedup (range 1.31–1.61× across tasks). At 25 ms, speedups rise to $\sim 2.19\times$ on average (range 1.90–2.45×), and at 50 ms they remain $\sim 2.04\times$ on average (range 1.80–2.26×). FluidPipe is also less latency-sensitive: as RTT grows from 0.01 ms to 50 ms, PP epochs slow by roughly 5.0–5.4×, whereas representative FP variants slow by only $\sim 3.7$–3.9×. In short, FP roughly halves the latency penalty while retaining its advantage at low RTT.

**Final accuracy.** Across CIFAR-100, Flowers-102, and Pets, all FP variants match or slightly exceed PP; notably, **FP-ND** (no cross-stage distillation) attains parity, which isolates the auxiliary head on Stage 1 as the main mechanism needed to preserve quality and remove the cross-stage gradients (Table 2). Distillation and the extra block are helpful but non-essential refinements: they stabilize and can yield small, task-dependent gains. Moreover, these two components are further discussed in Appendix A.

Table 3: CIFAR-100: best accuracy and runtime at best step vs PEFT (LoRA).

|  | FP-DT-EL | FP-ND-EL | PP | FP-DT | FP-DB | FP-ND | LoRA |
|---|---|---|---|---|---|---|---|
| Acc (%) | **93.54**±0.08 | 93.41±0.14 | 93.37±0.24 | 93.30±0.08 | 93.25±0.18 | 93.21±0.11 | 92.64±0.16 |
| Runtime (h) | 2.94±0.77 | **1.35**±0.39 | 2.69±0.83 | 1.59±0.52 | 2.09±0.29 | 1.84±0.52 | 7.52±3.13 |

Table 4: Mean per-epoch runtime (minutes): shown over multiple tasks and latencies. Speedup over PP is shown in parentheses.

| Task | Ag News | | IMDB | | Yelp Review Full | |
|---|---|---|---|---|---|---|
| Latency | 0.01 ms | 25 ms | 0.01 ms | 25 ms | 0.01 ms | 25 ms |
| Method | | | | | | |
| PP | 96.15 | 303.96 | 19.33 | 59.48 | 516.91 | 1707.26 |
| FP-DB | 86.2 (×1.12) | 91.13 (×3.34) | **15.01** (×1.29) | **19.57** (×3.04) | 470.74 (×1.1) | 553.81 (×3.08) |
| FP-DB-EL | 91.08 (×1.06) | 96.29 (×3.16) | 16.1 (×1.2) | 20.99 (×2.83) | 498.14 (×1.04) | 586.06 (×2.91) |
| FP-DT | **85.95** (×1.12) | **90.87** (×3.34) | 15.04 (×1.29) | 19.61 (×3.03) | 468.61 (×1.1) | 551.31 (×3.1) |
| FP-DT-EL | 90.9 (×1.06) | 96.11 (×3.16) | 16.07 (×1.2) | 20.95 (×2.84) | 494.86 (×1.04) | 582.2 (×2.93) |
| FP-ND | 86.03 (×1.12) | 90.96 (×3.34) | 15.05 (×1.28) | 19.62 (×3.03) | **440.77** (×1.17) | **518.56** (×3.29) |
| FP-ND-EL | 90.48 (×1.06) | 95.66 (×3.18) | 16.04 (×1.21) | 20.91 (×2.84) | 464.53 (×1.11) | 546.51 (×3.12) |

**Distributed Training vs. Parameter-Efficient Fine-Tuning (PEFT).** Table 3 contrasts distributed methods (FP and PP) with LoRA. We find that LoRA achieves comparable accuracy (92.64%) but lags behind distributed methods by nearly a percentage point, while requiring substantially longer runtime ($7.52 \pm 3.13$ h). In contrast, FluidPipe and PP consistently exceed 93% accuracy, with FluidPipe variants such as FP-ND-EL reaching 93.41% in only $1.35 \pm 0.39$ h. This highlights a key trade-off: PEFT reduces the number of trainable parameters but can still incur long wall-clock times due to sequential execution, whereas distributed training methods like FluidPipe shorten runtime dramatically without sacrificing accuracy. For practitioners with multi-GPU resources, FluidPipe therefore offers a more efficient path to high accuracy, while LoRA remains attractive in strictly single-GPU or memory-constrained settings.

## 5.2 SETUP B: BERT-LARGE FINE-TUNING ON TWO GPUS

We fine-tune BERT-LARGE (Devlin et al., 2019) on three text classification tasks (AG News, Yelp Review Full (Zhang et al., 2015), and IMDB Reviews (Maas et al., 2011)). We follow a standard HuggingFace fine-tuning recipe and evaluate *at the end of each epoch*. We use two machines (one with a GPU each): Stage 1 runs on node A and Stage 2 on node B. Since we have only 2 GPUs in total, we only change the *inter-node* algorithm (PP vs. FluidPipe variants).

**Epoch-Time Results.** Table 4 shows that FP again lowers epoch time and dampens latency effects. At 25 ms, AG News drops from 303.96 min (PP) to 90.87–96.29 min (FP)—3.16–3.34× faster; IMDB from 59.48 min to 19.57–20.99 min—2.83–3.04×; and Yelp from 1707.26 min to 518.56–586.06 min—2.91–3.29×. At 0.01 ms, FP still helps, with 1.06–1.29× speedups across tasks (e.g., IMDB: 19.33→15.01 min). Latency sensitivity starkly differs: PP slows by 3.08–3.30× moving from 0.01 ms to 25 ms , while representative FP runs grow by only $\sim 1.06$–1.30× . Thus, FP delivers ≈!3× speedups at realistic cross-zone RTTs and remains beneficial even at datacenter latency.

**Final Accuracy.** Table 5 shows that FluidPipe matches PP on AG News (92.75%), surpasses PP on IMDB (89.88% vs. 88.86%), and trails by less than 1 pt on Yelp (64.94% vs. 65.82%). Unlike Setup A, here we fix a small epoch budget; because FP completes each epoch much faster, a wall-clock–matched comparison would allow more FP epochs and could close the small Yelp gap. Figure 4 shows the accuracy over runtime at 0.01 ms RTT and Figure 5 shows it at 25 ms.

Overall, our results indicate that FluidPipe's central contribution is *eliminating per-iteration cross-stage gradient exchange* by equipping Stage 1 with an auxiliary head. This change reduces synchronization and communication while keeping iteration-time updates local—thereby improving efficiency—yet it preserves (and sometimes improves) end-to-end accuracy across setups. Look-

Table 5: Best accuracy (mean ± std) across datasets

|            | Yelp Review Full | AG News | IMDB |
|------------|------------------|---------|------|
| FP-DB      | $63.00_{\pm 0.29}\%$ | $92.10_{\pm 0.20}\%$ | $87.98_{\pm 2.80}\%$ |
| FP-DB-EL   | $62.82_{\pm 0.41}\%$ | $92.04_{\pm 0.21}\%$ | $87.74_{\pm 2.66}\%$ |
| FP-DT      | $64.86_{\pm 0.41}\%$ | $92.54_{\pm 0.29}\%$ | $89.79_{\pm 0.97}\%$ |
| FP-DT-EL   | $64.73_{\pm 0.61}\%$ | $92.56_{\pm 0.13}\%$ | $\mathbf{89.88}_{\pm 1.13}\%$ |
| FP-ND      | $64.94_{\pm 0.42}\%$ | $92.51_{\pm 0.11}\%$ | $89.74_{\pm 0.73}\%$ |
| FP-ND-EL   | $64.85_{\pm 0.60}\%$ | $\mathbf{92.75}_{\pm 0.07}\%$ | $89.51_{\pm 1.16}\%$ |
| PP         | $\mathbf{65.82}_{\pm 0.23}\%$ | $92.67_{\pm 0.24}\%$ | $88.86_{\pm 3.40}\%$ |

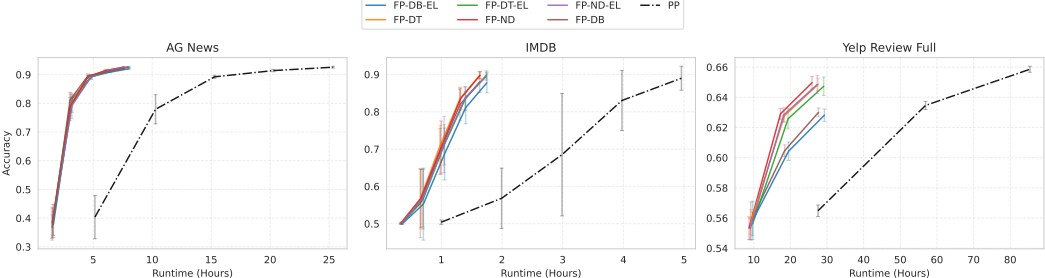

Figure 4: Accuracy vs runtime at $0.01\,\mathrm{ms}$ for AG News & IMDB & Yelp Review Full.

ing ahead, a key open question is whether auxiliary heads can simultaneously sustain quality and efficiency when placed on *multiple* intermediate stages in deeper pipelines.

## 6  CONCLUSION

We presented FluidPipe, a pipeline training algorithm that removes per-iteration cross-stage gradient synchronization by equipping the first stage with an auxiliary head and replacing gradients with coarse, low-frequency feedback. This shift keeps updates local, reduces latency sensitivity, and consistently accelerates training while preserving accuracy. Our analysis explains why the dependency change yields speedups, and our experiments on ViT-Large and BERT-Large confirm gains across both datacenter and cross-region latencies. Distillation is optional, highlighting that the auxiliary head itself is the key enabler. While we focused on the two-stage case to isolate the mechanism, extending FluidPipe to deeper pipelines raises a central open question: can intermediate stages trained under auxiliary heads retain sufficient capacity and align with the global task? Addressing this challenge is essential for realizing the full potential of gradient-free pipelining.

**Ethics Statement.**  This work focuses on distributed training algorithms for large language and vision models. Our study does not involve human subjects, personally identifiable information, or sensitive data. We rely exclusively on publicly available benchmark datasets (CIFAR-100, Oxford-

Figure 5: Accuracy vs runtime at $25\,\mathrm{ms}$ for AG News & IMDB & Yelp Review Full.

IIIT Pets, Flowers-102, AG News, Yelp, IMDB) that are widely used in the machine learning community. Our methods are intended to improve the efficiency of fine-tuning large models across multi-GPU and multi-node systems; they do not generate new content or decision policies that could directly impact individuals. We acknowledge that improved efficiency in training large models may indirectly lower barriers to training, which could accelerate both beneficial and potentially harmful applications of foundation models. We believe that these broader ethical considerations warrant ongoing discussion in the community, but our contribution is purely methodological and system-level, with no direct risks of misuse beyond those already inherent in large-scale ML.

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

## A EXPERIMENTS & RESULTS

**Reproducibility Statement.** We have taken multiple steps to ensure reproducibility of our results. Our paper provides a full description of the FluidPipe algorithm (Section 3), a cost and communication model with all assumptions stated (Section 4), and detailed experimental setups including model architectures, datasets, hardware environments, and latency emulation methods (Section 5). We report results across three random seeds for each configuration and include standard deviations in accuracy tables. We will release anonymized source code and configuration files as supplementary material to enable independent verification of our results and reproducibility.

### A.1 ADDITIONAL EXPERIMENTS: VIT-BASE

To further validate the design choices in FluidPipe, we conduct a set of experiments using ViT-Base on CIFAR-100, which allows faster iteration while preserving the key behaviors of interest. Specifically, we investigate:

- the role of distillation and how to weight it effectively,
- the impact of different split points on model accuracy, and
- the utility of the extra block as a design enhancement.

**Exploring Distillation.** We vary $\alpha$, the weight between the task loss and the distillation loss:

$$\mathcal{L}_{total} = \alpha\mathcal{L}_{task} + (1 - \alpha)\mathcal{L}_{distillation}. \tag{1}$$

A higher $\alpha$ places more weight on the task loss; $\alpha = 1$ corresponds to no distillation. Since $\mathcal{L}_{total}$ is computed at both stages, we evaluate a grid of $\alpha$ values for the partial (stage 1) and full (stage 2) models.

Figure 6 reports the best accuracy of the full model and the corresponding epoch. The highest accuracies (around $92.3\%$) are achieved across a range of settings where the full model is distilled

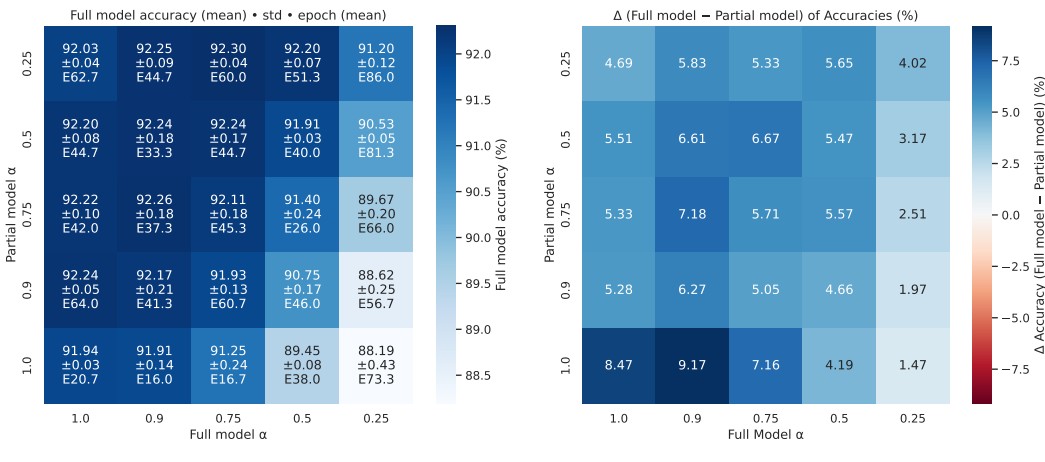

(a) Full model best accuracy      (b) $\Delta$ (full model−partial model) best accuracy

Figure 6: Alpha grid: full model accuracy and $\Delta$ between FP full model vs partial model across $\alpha$ values.

Table 6: Effect of number of blocks per FluidPipe stage and the addition of Extra Block at M1

| | w/o extra block | | | w extra block | | |
|---|---|---|---|---|---|---|
| Split | Accuracy (mean ± std) | Epoch @ best (mean) | Runtime | Accuracy | Epoch | Runtime |
| (10, 4) | 91.96 ± 0.03% | 16.0 | 0.87 h | 92.18 ± 0.12% | 35.3 | 2.23 h |
| (12, 2) | 92.14 ± 0.06% | 20.7 | 1.35 h | 92.20 ± 0.08% | 16.7 | 1.23 h |
| (2, 12) | 86.63 ± 0.10% | 79.3 | 5.64 h | 91.55 ± 0.10% | 42.7 | 2.91 h |
| (4, 10) | 90.94 ± 0.23% | 14.0 | 0.82 h | 91.72 ± 0.16% | 77.3 | 4.57 h |
| (7, 7) | 91.46 ± 0.15% | 12.7 | 0.58 h | 91.95 ± 0.07% | 24.0 | 1.16 h |

lightly (higher $\alpha$) while the partial model receives moderate distillation (lower $\alpha$). This supports our intuition: **distillation helps the partial model learn in place of receiving gradients**. We also note that full-model accuracy is relatively stable across many $\alpha$ combinations once distillation is tuned, with differences of less than 1% between most settings.

Furthermore, Figure 6 shows that convergence speed varies sharply with $\alpha$. The fastest convergence (16–21 epochs) occurs when the partial model is trained without distillation ($\alpha = 1$) and the full model receives minimal to no distillation ($\alpha \approx 0.75$–1.0). In contrast, when both stages are distilled heavily ($\alpha \leq 0.5$), convergence is substantially slower, with the best accuracy only appearing after 40–80 epochs.

We also plot $\Delta$ accuracy, defined as the difference between the full and partial model accuracies. This serves as a sanity check (the full model should outperform the partial) and shows how distillation changes the gap. For example, $\Delta$ shrinks from about 9% when the full model is not distilled at all ($\alpha = 1$) to around 2% when it is distilled heavily ($\alpha = 0.25$). This shrinkage, however, reflects degraded full-model accuracy rather than genuine improvement of the partial model.

> **Practitioner Takeaway**
>
> A practical rule of thumb is to distill more to the partial model while keeping the full model mostly task-oriented.

**Split Point and Extra Block.** Figure 7 reports accuracy for both the partial model (stage 1) and the full model (stage 2) across split points, with and without the extra block; Table 6 reports full-model accuracy (mean ± std), as well as epochs and runtime to best.

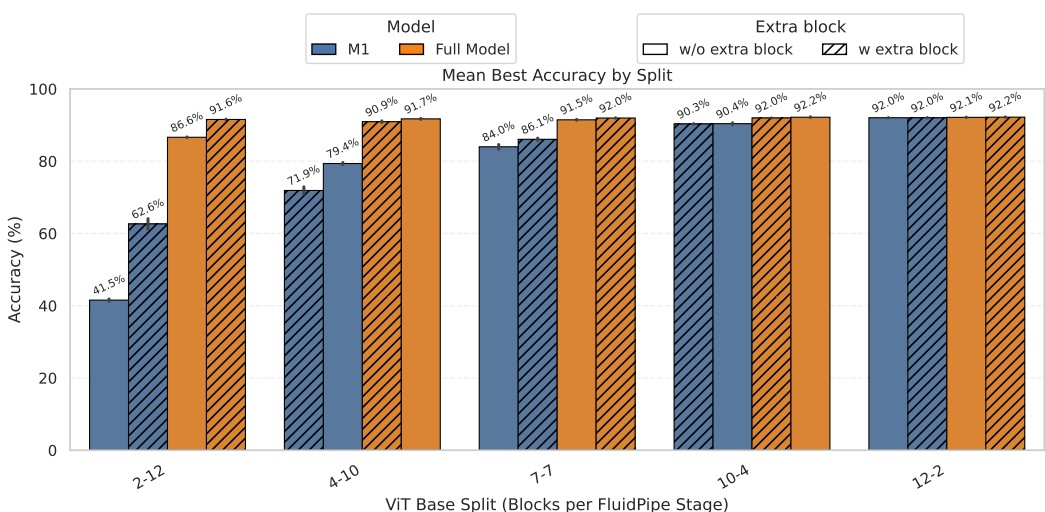

Figure 7: Split and Extra Block utility.

*Split point (no extra block).* For the full model, accuracy is robust under balanced and moderately skewed splits: (7,7) $91.46\%$, (10,4) $91.96\%$, and (12,2) $92.14\%$ (Table 6). Only the extreme (2,12) split collapses accuracy to $86.63\%$. In contrast, the figure shows that the partial model depends strongly on stage 1 depth: it is high when stage 1 is deep (e.g., (12,2) $\approx 92.0\%$) and very low when stage 1 is shallow (e.g., (2,12) $\approx 41.6\%$).

*Balanced split with extra block (7,7).* Adding the extra block increases full-model accuracy from $91.46\% \to 91.95\%$ ($+0.49$ pp; Table 6), and improves the partial model from $\approx 84.0\% \to 86.1\%$ (Figure 7). This comes with slower convergence for the full model ($12.7 \to 24.0$ epochs; $0.58 \to 1.16$ h).

*Interaction.* Across splits, the extra block consistently raises accuracy for both models. For the full model, gains are modest under balanced/moderate splits—(10,4): $91.96\% \to 92.18\%$, (12,2): $92.14\% \to 92.20\%$, (7,7): $91.46\% \to 91.95\%$—but are substantial when stage 1 is under-provisioned: (2,12): $86.63\% \to 91.55\%$ ($+4.92$ pp) and (4,10): $90.94\% \to 91.72\%$ ($+0.78$ pp) (Table 6). The figure shows an even larger effect on the partial model when stage 1 is shallow: (2,12): $\approx 41.6\% \to 62.6\%$ ($+21.0$ pp), (4,10): $\approx 71.9\% \to 79.4\%$ ($+7.5$ pp). Effects on convergence vary by split: the extra block can slow training (e.g., (4,10): $14.0 \to 77.3$ epochs) or speed it up (e.g., (12,2): $20.7 \to 16.7$ epochs).

*Takeaways.* (i) Full-model accuracy is robust to the split point except in the extreme (2,12) case; (ii) the extra block is *useful in general*, providing small but consistent gains (e.g., $+0.49$ pp at (7,7)); and (iii) the extra block is *especially* beneficial when stage 1 is shallow, where it substantially lifts both the partial and full model accuracy.

## A.2   ADDITIONAL EXPERIMENTS: BERT-LARGE

Following Setup B, we probe the effect of distillation on IMDB in Figure 8. In contrast to Figure 6 (ViT), we observe little sensitivity to the choice of Stage 1 (partial model) $\alpha$ over a broad range, and accuracy *degrades* when distillation into the full model (Stage 2) is strong (i.e., $0.5$–$0.25$ $\alpha_2$). A plausible explanation is the short training budget in this setup: we run only 5 epochs. As seen in Figure 6, the configurations that benefited from distillation typically reached their best accuracy later in training; with a small fixed epoch budget, that advantage does not have time to materialize.

*Practical takeaway.* Under small epoch budgets, prefer little-to-no distillation into Stage 2 (e.g., $\alpha_2 \approx 0.75$–$1.0$); heavier Stage 2 distillation is counterproductive, while tuning $\alpha_1$ has comparatively minor effect in this regime.

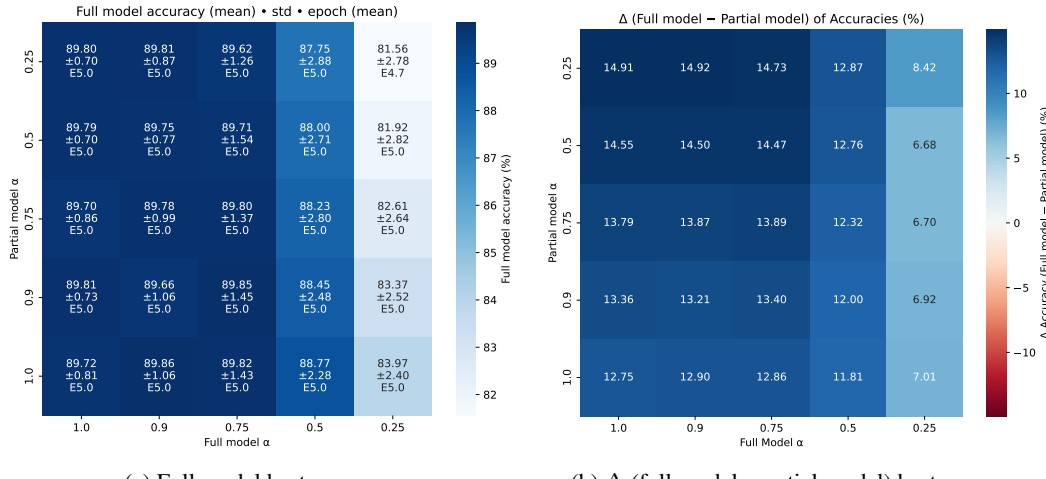

(a) Full model best accuracy

(b) $\Delta$ (full model$-$partial model) best accuracy

Figure 8: Alpha grid: full model accuracy and $\Delta$ between FP full model vs partial model across $\alpha$ values.

### A.3 ADDITIONAL EXPERIMENTS: FP VS SoTA PIPELINE SCHEDULERS

We execute a small experiment to compare the training time Figure 9, idle time of stage 1 Figure 10, and idle time of stage 2 Figure 11 of FluidPipe and different Pipeline schedulers [GPipe/1F1B/zero-bubble Huang et al. (2019); Narayanan et al. (2019); Qi et al. (2023)]. We used ViT-Base on two nodes with 1 A100 each, full precision training on a dummy images (224x224x3) dataset. We used a fixed batch size of 256 and varied the number of micro-batches.

FP surpasses GPipe, 1F1B and Zero-Bubble train time even at 0 ms RTT (100 Gbps), where scheduling methods are most effective. Moreover, FP removes the idle times caused by pipeline bubbles. Scheduling and FP operate at different layers of abstraction: scheduling controls micro-batch interleaving within a stage, whereas FP removes the backward-path dependency between stages.

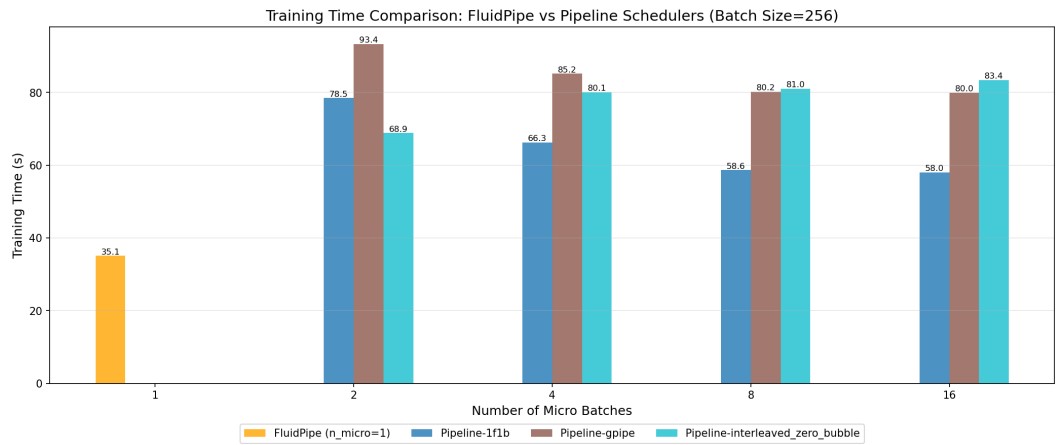

Figure 9: The training time of FP vs PP with different schedulers.

### A.4 COMPUTATIONAL TIMELINE

Figure 12 shows a computational timeline of the two stages during the execution of an epoch using FP and PP. This result confirms that FP removes the dependency of stage 1 on stage 2, that is, stage 1 finishes the computations faster. Moreover, we can see that there are fewer bubbles between computations. An interesting future direction is to utilize the idle time at stage 2.

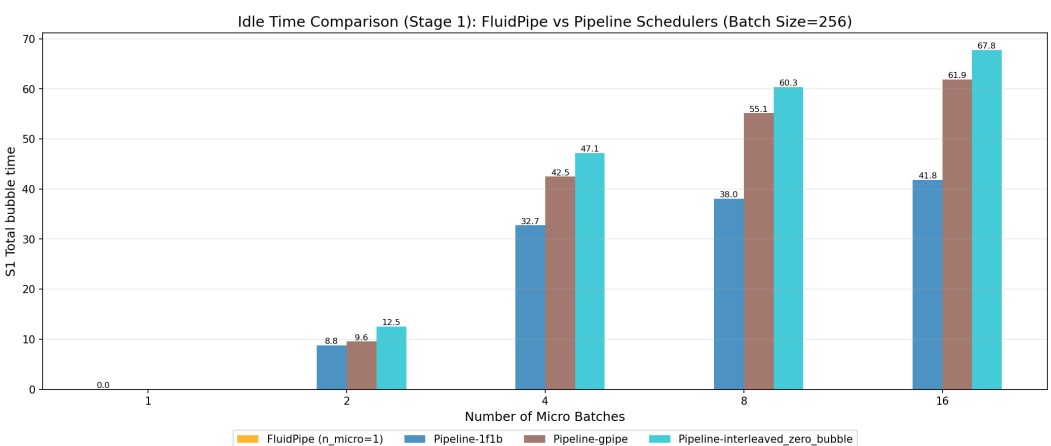

Figure 10: The bubble (idle) time of Stage 1 of both PP and FPs.

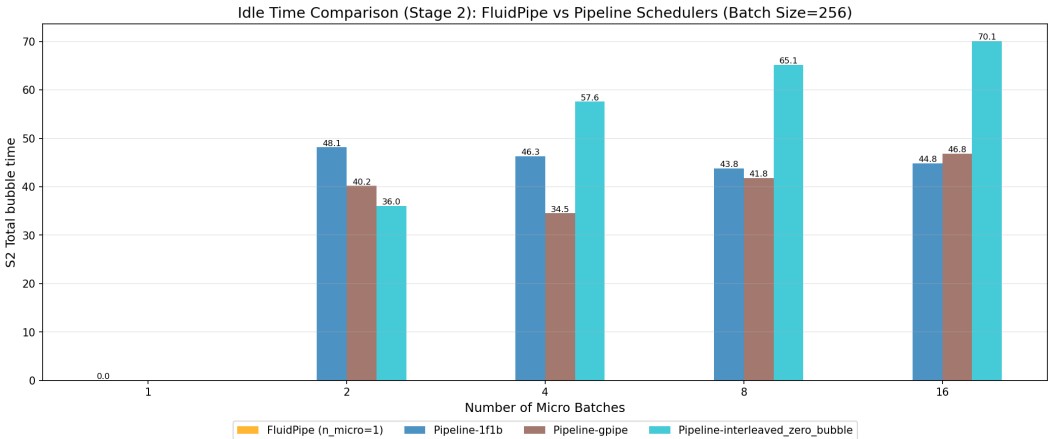

Figure 11: The bubble (idle) time of Stage 2 of both PP and FPs.

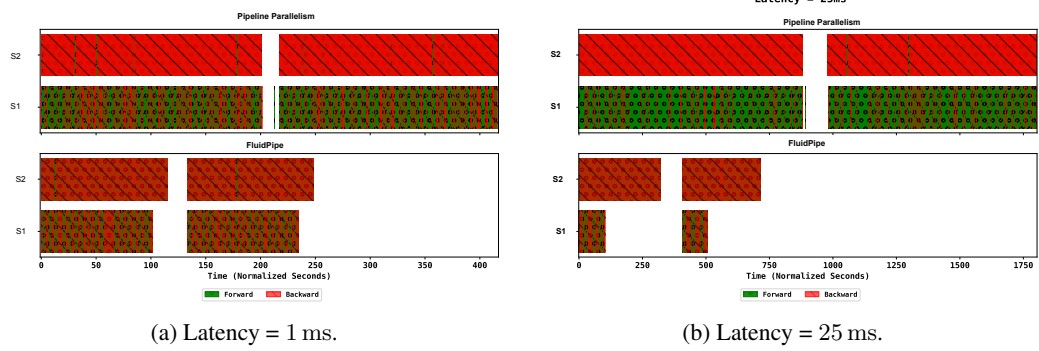

(a) Latency = 1 ms.

(b) Latency = 25 ms.

Figure 12: Iteration timeline comparing Pipeline Parallelism and FluidPipe. This plot shows the computation performed on two stages during two epochs.

