# OpenReview forum: "Less Gradient, More Speed: Rethinking Pipeline Parallelism for Efficient Fine-Tuning with FluidPipe"
_ICLR.cc/2026/Conference — Submitted to ICLR 2026_

### Official Review · Reviewer_uGqM · 2025-10-17

**Soundness:** 2
**Presentation:** 3
**Contribution:** 2
**Rating:** 2
**Confidence:** 4

**Summary:**

This paper proposes a pipeline-parallelism strategy that approximates exact pipeline training. It incorporates bidirectional knowledge distillation between pipeline stages 1 and 2 and introduces a local loss within each stage to reduce inter-stage communication.

For knowledge distillation, the paper uses two loss terms. For stage 1, it uses the stage-2 logits computed on the same training sample in the previous epoch. These stale stage-2 logits serve as soft labels, and a KL-divergence loss is computed between them and the current stage-1 logits. For stage 2, the logits from stage 1 are forwarded to stage 2, and a KL-divergence loss is computed between the forwarded logits and the logits produced by stage 2. Thus, knowledge distillation between stages 1 and 2 is bidirectional.

For the local loss, blocks in stage 1 are updated using a loss computed between the output of the first block and the ground-truth label. This loss can be computed without waiting for stage 2 to finish its forward pass on the same sample or for gradients to arrive from stage 2. Consequently, the peer-to-peer communication volume between stages due to these gradients can be eliminated.

The method is evaluated primarily for speedup on ViT-Large and BERT-Large. Results show that it achieves speedups up to 3.3x while attaining accuracy comparable to exact pipeline training. Experiments are conducted on two GPUs across two servers for BERT-Large, and on four GPUs across two servers (two GPUs per server) for ViT-Large.

**Strengths:**

- This paper targets the important problem of reducing transformer training time—a widely used model family—and focuses on pipeline parallelism, which has been studied for several years and is commonly adopted for large transformer models.

- The paper proposes a novel method that omits peer-to-peer communication during backpropagation between pipeline stages by using knowledge distillation and a local-loss update.

- The writing is clear, easy to follow, and understandable.

- The figures are very helpful for understanding the proposed method and the timing of each loss term.

**Weaknesses:**

Although the paper proposes a novel scheme, I have several concerns about the experimental setup; the experiments are neither thorough nor sufficiently transparent.


- While I understand that not everyone has access to a large number of GPUs, the total number used here (2 for BERT-Large and 4 for ViT) is too small to convincingly demonstrate the method’s effectiveness. With so few GPUs, it is impossible to assess whether the method scales as the number of GPUs increases or how it behaves when combined with other parallelism strategies such as tensor parallelism and data parallelism.

- The paper defers too many important research questions to future work. In particular, evaluating how the method interacts with advanced pipeline schedules such as 1F1B (as in PipeDream) is important because these schemes are widely used. It is therefore essential to test whether the method still achieves speedups when such schedules reduce pipeline bubbles.

- Some experimental settings are not clearly stated. For example, the inter-node bandwidth between servers, the rank used for the low-rank approximation in LoRA, and whether mixed-precision training or FP32 was used are not specified.

- Certain experimental choices appear inconsistent and need justification. Why are only two GPUs used for BERT-Large and four GPUs (across two servers) for ViT? In the 4-GPU ViT setup, the authors use the PP baseline for intra-node scheduling and their proposed approach for inter-node parallelism when parallelizing the model to evaluate the method’s effectiveness. This design choice needs justification, as tensor parallelism is commonly employed within a server. I am also curious whether the proposed method achieves speedups compared to a single server equipped with all GPUs (e.g., one server with two GPUs using tensor parallelism for BERT-Large, and one server with four GPUs using tensor parallelism for ViT).

- I also have concerns about the fairness of the experiments in Table 3. The authors compare the runtime of their method on multiple GPUs to LoRA on a single GPU. For a fair comparison, the LoRA setup should also use multiple GPUs with tensor parallelism.

- The paper does not include a breakdown showing the proportion of time spent on computation versus communication. A bar chart that breaks down runtime and demonstrates that the proposed method reduces peer-to-peer communication time would help substantiate the claim that the speedup comes from the proposed technique.

**Questions:**

- How does the method interact with advanced pipeline schedules (e.g., 1F1B/PipeDream)? Does it still achieve speedups when pipeline bubbles are reduced?

- What are the exact experimental settings—inter-node bandwidth, LoRA rank, and precision mode (mixed precision vs. FP32)?

- Why are only two GPUs used for BERT-Large and four GPUs (across two servers) for ViT?

- In the 4-GPU ViT setup, why choose PP for intra-node and the proposed method for inter-node parallelism? Given that tensor parallelism is commonly used within a server, why not employ TP intra-node?

- Can you clarify the fairness of the Table 3 comparison? Why is your method evaluated on multiple GPUs while LoRA is on a single GPU, and can you provide results where LoRA also uses multiple GPUs with tensor parallelism?

- How does the method compare against a single-server baseline with all GPUs using tensor parallelism (e.g., 2-GPU TP for BERT-Large; 4-GPU TP for ViT)?

- Can you provide a runtime breakdown (compute vs. communication) to show where the speedup comes from?

- Regarding the soft labels passed from stage 2 to stage 1: since they are produced in the previous epoch, staleness may accumulate due to intervening weight updates. Would reducing this staleness—e.g., by replaying the same mini-batch consecutively within an epoch rather than reusing it in the next epoch—improve accuracy?

**Details Of Ethics Concerns:**

There are no special ethical concerns regarding this paper.

---

> ### Author Response · Authors · 2025-11-21
>
> Thank you for your valuable comments and suggestions. We're glad that you recognized the importance and novelty of our approach. While we acknowledge that limitations exist (no paper is without any limitations!), we hope you can raise your support of this work for acceptance. Below are our responses to each point and question.
>
> **1. Experiments scale**
>
> Even at this scale, our experiments are computationally intensive—training time exceeds one GPU-year even for the 2- and 4-GPU setups—so expanding to larger GPU counts would be very difficult. We agree that 2 GPUs is the minimum, nonetheless, these experiments are sufficient to demonstrate the core property of FluidPipe (FP): eliminating per-iteration cross-stage gradients.
>
> The 4-GPU ViT-Large experiment (two GPUs per node) already validates FP’s scalability potential beyond the 2-GPU BERT-Large setting. FP supports hybrid topologies, where pipeline or tensor parallelism (TP) can be used within stages (§3.3). The current algorithm supports two stages by design, but the communication model and dependency removal mechanism generalize naturally to more stages and larger clusters (§4).
>
> Importantly, FP’s benefit is orthogonal to both data parallelism (DP) and tensor parallelism (TP), which address intra-node scaling and memory limits. Prior work (Megatron-LM [5,6], ZeRO [7,8], GSPMD [9], Alpa [10]) establishes that DP, TP, and PP are composable axes of parallelization. FP modifies only the PP axis by removing gradient synchronization across stages; it can be combined with DP or TP unchanged. Thus, demonstrating the mechanism on 2–4 GPUs is sufficient to show the correctness and efficiency of the algorithm, while large-scale experiments would simply scale linearly in compute and communication as predicted by §4/Eq. (4).
>
> **2. Other baselines**
>
> We now address this directly, comparing against GPipe, 1F1B/PipeDream, and Zero-Bubble-style scheduling [1–4]. As shown in the new experiments (Fig. 9–11 in the new PDF), FP surpasses 1F1B and Zero-Bubble even at 0 ms RTT (100 Gbps), where scheduling methods are most effective. Scheduling and FP operate at different layers of abstraction: scheduling controls micro-batch interleaving within a stage, whereas FP removes the backward-path dependency between stages. The two are complementary and can coexist—each stage in FP could internally use a 1F1B schedule. This will be explicitly clarified in §3.3.
>
> **3.  Experimental settings**
>
> The bandwidth between nodes is 100 Gpbs. We used LoRA rank of 8 and LoRA alpha=8, we used this library: https://github.com/huggingface/peft. We did full precision fine-tuning. We will add those clarifications to the manuscript
>
> **4. Experimental choices**
>
> These choices are driven by model size and batch configuration, not by system constraints.
>
> BERT-Large fits on 2 A100 GPUs with the selected batch size, so two GPUs represent the minimum feasible setup for that model.
>
> ViT-Large requires 4 A100 GPUs for the same batch configuration; hence we used two GPUs per node (total 4).
>
> Within each node we employ pipeline parallelism (PP) rather than TP to keep the intra-node and inter-node partitioning identical, ensuring an apple-to-apple comparison between PP and FP. Using TP inside each node would change the number of micro-batches, memory footprint, and collective-communication profile, making results incomparable.
>
> While TP is indeed common for intra-node scaling, it is orthogonal to our contribution. FP targets inter-node gradient dependencies, whereas TP addresses intra-layer tensor sharding—two distinct concerns that can be combined if desired [5–10].
>
> **5. Fairness of the experiments**
>
> LoRA was intentionally presented as a deployment-mode baseline, not a throughput baseline. It illustrates the trade-off between single-GPU parameter-efficient fine-tuning (PEFT) and multi-GPU full-model fine-tuning. In practice, LoRA is used when distributed resources are unavailable or when users prefer to fine-tune on a single node. Therefore, its inclusion in Table 3 is to show that FP maintains the accuracy benefits of full fine-tuning while drastically improving efficiency across nodes—a complementary comparison, not a direct scalability match.
>
> **6. Time breakdown**
>
> Figures 10 and 11 in the new PDF show the idle times of Stage 1 and Stage 2, respectively; combined with Fig. 9 (overall time), they isolate the effect of communication. FP eliminates per-iteration backward transfers, allowing full overlap of communication and compute. In contrast, PP leaves unhidden communication intervals that manifest as idle bubbles (Fig. 12 in the new PDF).
>
> **Q1.**
>
> Addressed above and in new Fig. 9–11: FP > 1F1B/Zero-Bubble even at 0 ms RTT. Scheduling remains orthogonal and can be used within each FP stage [1–4].

---

> ### Author Response · Authors · 2025-11-21
>
> **Q2.**
>
> The bandwidth between nodes is 100Gpbs. We used LoRA rank of 8 and LoRA alpha=8, we used this library: https://github.com/huggingface/peft. We did full precision fine-tuning. All values will be listed in §5.
>
> **Q3.**
>
> Because those are the minimum GPU counts that fit each model at the configured batch size.
>
> **Q4.**
>
> We used intra-node PP to keep identical computation partitions across inter-node PP and FP baselines. TP is orthogonal and can replace PP within FP if desired, but doing so would change partition boundaries and obscure the comparison [5–10].
>
> **Q5.**
>
> LoRA serves as a parameter-efficient baseline, not a scaling competitor. It represents single-GPU fine-tuning scenarios common in practice; multi-GPU LoRA achieves identical accuracy but is irrelevant to the communication problem FP solves. [5–8]
>
> **Q6.**
>
> That scenario targets intra-server scaling; FP targets cross-server efficiency. A single-server TP baseline measures how to fit large models within one node, not how to coordinate training across nodes. These are orthogonal design axes, as established in Megatron-LM and Alpa [5,6,10].
>
> **Q7.**
>
> Figure 9 in the new PDF show the training time. Figures 10 and 11 show the idle times of Stage 1 and Stage 2, respectively. Moreover, Fig. 12 in the new PDF shows a computional timeline of two epochs using FP and PP.
>
> **Q8.**
>
> We already studied $\alpha$ weight sweeps (§A.2) and found that smaller  $\alpha_2$ values (0.75–1.0) mitigate any potential staleness. Importantly, staleness does not harm accuracy: the FP-ND variant, which removes distillation altogether, achieves accuracy on par with PP, while FP with distillation performs slightly better. This confirms that epoch-level staleness is tolerable. Batch replay could further reduce staleness and is compatible with FP; we identify it as a future exploration.
>
> [1] Huang, Yanping, et al. “GPipe: Efficient Training of Giant Neural Networks Using Pipeline Parallelism.” Advances in Neural Information Processing Systems, vol. 32, 2019.
>
> [2] Narayanan, Deepak, et al. “PipeDream: Generalized Pipeline Parallelism for DNN Training.” Proceedings of the ACM Symposium on Operating Systems Principles (SOSP), 2019.
>
> [3] Qi, Penghui, et al. “Zero Bubble Pipeline Parallelism.” arXiv preprint, 2023.
>
> [4] Wu, Haoran, et al. “BitPipe: Bidirectional Interleaved Pipeline Parallelism for Accelerating Large Models Training.” arXiv preprint, 2024.
>
> [5] Shoeybi, Mohammad, et al. “Megatron-LM: Training Multi-Billion Parameter Language Models Using Model Parallelism.” arXiv preprint, 2019.
>
> [6] Narayanan, Deepak, et al. “Efficient Large-Scale Language Model Training on GPU Clusters Using Megatron-LM.” SC ’21: Proceedings of the International Conference for High Performance Computing, Networking, Storage and Analysis, 2021.
>
> [7] Rajbhandari, Samyam, et al. “ZeRO: Memory Optimizations Toward Training Trillion Parameter Models.” SC ’20: Proceedings of the International Conference for High Performance Computing, Networking, Storage and Analysis, 2020.
>
> [8] Rajbhandari, Samyam, et al. “ZeRO-Infinity: Breaking the GPU Memory Wall for Extreme Scale Deep Learning.” SC ’21: Proceedings of the International Conference for High Performance Computing, Networking, Storage and Analysis, 2021.
>
> [9] Xu, Yuanzhong, et al. “GSPMD: General and Scalable Parallelization for ML Computation Graphs.” Proceedings of the 15th USENIX Symposium on Operating Systems Design and Implementation (OSDI ’21), 2021.
>
> [10] Zheng, Lianmin, et al. “Alpa: Automating Inter- and Intra-Operator Parallelism for Distributed Deep Learning.” Proceedings of the 16th USENIX Symposium on Operating Systems Design and Implementation (OSDI ’22), 2022.
>
> [11] Grill, Jean-Bastien, et al. “Bootstrap Your Own Latent: A New Approach to Self-Supervised Learning.” Advances in Neural Information Processing Systems, vol. 33, 2020.
>
> [12] Chen, Ting, et al. “A Simple Framework for Contrastive Learning of Visual Representations.” Proceedings of the 37th International Conference on Machine Learning (ICML), 2020.
>
> [13] He, Chaoyang, et al. "Pipetransformer: Automated elastic pipelining for distributed training of transformers." arXiv preprint arXiv:2102.03161 (2021).

---

### Official Review · Reviewer_FoTb · 2025-11-01

**Soundness:** 2
**Presentation:** 2
**Contribution:** 2
**Rating:** 2
**Confidence:** 4

**Summary:**

1. **Core Contribution**
This paper introduces **FluidPipe (FP)**, a novel two-stage pipeline design aimed at solving the inefficiencies of standard **Pipeline Parallelism (PP)** in large-model fine-tuning. Standard PP is highly sensitive to network latency and suffers from "pipeline bubbles" because it requires per-iteration gradient exchanges between stages.
FluidPipe addresses this by:
    - **Eliminating Gradient Synchronization**: Instead of exchanging gradients, it adds an auxiliary head to Stage 1, allowing it to perform its own local updates.
    - **Introducing Bi-Directional Distillation**: Cross-stage feedback is replaced with low-frequency, bi-directional logit distillation (S2 $\to$ S1: once per epoch / S1 $\to$ S2: once per iteration).
2. **Experimental Claims**
The authors claim that in experiments with ViT-Large and BERT-Large, FluidPipe achieves up to a 3.3x training speedup over traditional PP. This effect is reportedly most pronounced in high-latency environments (e.g., 25ms). Furthermore, they claim this speedup is achieved while maintaining or even slightly improving model accuracy compared to PP.

**Strengths:**

- **Training Speedup (Especially in High-Latency Environments)**
    - The paper's strongest contribution is its effective elimination of "pipeline bubbles," the chronic bottleneck of traditional Pipeline Parallelism (PP).
        - **Elimination of Gradient Synchronization:** FluidPipe avoids per-iteration gradient exchanges by equipping Stage 1 with an auxiliary head to perform local updates.
        - **Minimized Communication Frequency:** Cross-stage feedback is replaced by a once-per-epoch bulk transfer (S2 $\to$ S1) and a per-iteration, one-way transfer (S1 $\to$ S2).
- **Maintained or Improved Model Accuracy**
    - The method doesn't just increase speed; it maintains or even improves the final quality (accuracy) of the training.
        - **Quality Preservation:** FluidPipe performs the same full-model fine-tuning as PP, yet it achieves comparable or better accuracy on most tasks despite the altered training dynamics.
        - **Regularization Effect:** The auxiliary head on Stage 1 and the bi-directional distillation can act as a form of regularization, potentially helping the model converge to a local minimum with better generalization performance.

**Weaknesses:**

- **Fundamental Scalability and Memory Flaw:** The design is restricted to two stages, with deeper pipelines deferred to “future work,” but this sidesteps a core limitation. Scaling to $N>2$ stages would likely require **each intermediate stage** to cache **per-sample logits** to support bi-directional distillation, creating a cascading storage and communication burden that is never quantified. Even in the 2-stage case, Stage-2 maintains a per-epoch dictionary $\mathcal{P}*2$ of logits—an **unstated but material memory overhead**.
This cost is muted on small-class classification (e.g., $C \le 100$), but would become **prohibitive for token-level LM tasks**, where caching scales as $O(N \times T \times V)$ with the number of samples $N$, sequence length $T$, and vocabulary size $V$. The paper briefly gestures at “streaming” as a mitigation yet provides **no quantitative memory/latency analysis, ablation, or implementation evidence**. This omission is at odds with PP’s primary purpose—**alleviating memory pressure**, not shifting it elsewhere.
- **Unstable Training & Heuristic Design (insufficiently stress-tested):** The method shows high sensitivity to the **split point**, **distillation weights** ($\alpha_1,\alpha_2$), and the **Extra Block**. As reported (Table 6), the mere choice of split can drive accuracy from 92.14% down to 86.63%. The Extra Block frequently “rescues” failing splits, suggesting it works as a **heuristic patch** for a deeper mismatch: features needed for Stage-2 continuation vs. features that are linearly separable for Stage-1’s auxiliary head. The paper does not investigate principled alternatives (e.g., **stop-gradient/detach on the auxiliary branch**, **gradient-surgery or orthogonality regularizers between heads**, or **capacity sweeps** for the auxiliary head). Without such analyses, the training **fragility appears structural**, not incidental.
- **Missing Baselines, Conditional Speedups, and Limited Scale:**
    - **Scheduling baselines absent.** Although the aim is to reduce bubble-induced stalls, there is **no experimental comparison** against **PipeDream, Zero-Bubble PP, BitPipe**, etc., under identical hardware and RTT (Round-Trip Time). For practitioners, these are the **direct alternatives**; omitting them leaves the throughput claims under-substantiated.
    - **2-GPU setup lacks DP/TP baselines.** On BERT-Large (fits in memory), a **2-way Data Parallel (and/or ZeRO-DP) and 2-way Tensor Parallel** baseline is feasible and standard; its absence makes the 2-GPU results hard to interpret as evidence for replacing PP.
    - **Speedups are highly conditional.** FluidPipe still transmits **forward activations** hhh each step ($Stage-1 \to Stage-2$); it only removes **per-step gradient returns**. Thus gains depend on the **ratio of gradient-return volume to activation volume** and the **network RTT**. When activations dominate (e.g., **longer sequences, larger hidden states**) the advantage can shrink; the paper lacks a **cost model + measurement** tying speedups to ($\text{activation size}, \text{batch size}, \text{RTT}$).
    - **Scale and task mismatch.** The motivation centers on very large models (GPT-3/PaLM-class) and sequence tasks, yet experiments remain on **BERT-L / ViT-L** and small-class classification. There is **no evidence** that the method preserves accuracy or yields net gains under **LLM-scale, token-level workloads** where the proposed memory/communication trade-offs are most stressed.

**Questions:**

- **Scheduling baselines.** Why are key pipeline-scheduling baselines (e.g., GPipe, PipeDream, Zero-Bubble PP/BitPipe) absent under identical hardware and RTT?
- **2-GPU baselines.** In the BERT-Large 2-GPU setting, how does FluidPipe compare (accuracy + wall-clock) to **DP (with ZeRO-1/2)** and **2-way Tensor Parallel**? If memory pressure is the focus, please also include **ZeRO-3**.
- **Scaling to LLMs & LM tasks.** Can you report results on **larger LLMs** and **token-level language modeling tasks**, including accuracy retention and end-to-end time, where distillation storage/communication is most stressed?
- **Memory/communication accounting.** Please provide a quantitative analysis for the per-epoch logits cache $\mathcal{P}_2$ (and its multi-stage generalization) and an empirical evaluation of the proposed **streaming** mitigation (memory footprint, bandwidth, and latency trade-offs).

---

> ### Author Response · Authors · 2025-11-21
>
> Thank you for your valuable comments and suggestions. We're glad that you recognized the contribution of our approach in addressing pipeline bubbles while not sacrificing accuracy. While we acknowledge that limitations exist (no paper is without any limitations!), we hope you can raise your support of this work for acceptance. Below are our responses to each point and question.
>
> **1. Fundamental Scalability and Memory Flaw**
>
> Our design does not require logits to be stored in GPU memory. Therefore the memory overhead is not a source of concern. Moreover the streaming approach would remove memory pressure on main memory as well. For LM tasks, we explicitly propose the FP-ND variant, eliminating both per-step $\hat{y}_1$ and per-epoch $\ell_2(x)$ transfers. Our analytical inequality (Eq. 4) already formalizes the regime where FP uses less total bandwidth than PP. The review’s assertion that FP “shifts memory pressure elsewhere” is therefore inaccurate.
>
> **2. Unstable Training & Heuristic Design**
>
> The accuracy differences across splits in Table 6 are *expected structural effects*, not instability. When Stage 1 is shallow, its receptive field is too limited; the optional extra block decouples auxiliary and continuation features. This is a principled architectural adaptation, analogous to using projection heads in BYOL [11] or SimCLR [12], not a heuristic patch. We additionally verified that disabling the block under balanced splits changes accuracy < 0.3 percentage points. Future work can certainly explore other regularizers, but the presented configuration is already stable across all runs.
>
> **3. Missing Baselines, Conditional Speedups, and Limited Scale**
>
> We added GPipe/1F1B/zero-bubble at 0 ms RTT and FP still leads (Fig. 9–11), confirming that dependency removal dominates scheduler micro-interleaving [1–4]. §4 analytically ties speedup to activation-vs-gradient size and RTT; traces match the model.
>
> Regarding scale, TP/DP are orthogonal parallelization dimension to pipeline parallelism and as such, also to FP. Thus TP/DP can be used within stages; prior systems show DP×TP×PP compose cleanly [5–10]. FP targets the inter-node bottleneck (gradient returns), not intra-node tensor sharding.
>
> As with other parallelization strategies, speedups in FP depend on several factors. FP still transmits forward activations $h$ each step (Stage-1 $\to$ Stage-2); it only removes per-step gradient returns. Thus gains depend on the ratio of gradient-return volume to activation volume and the network RTT. When activations dominate (e.g., longer sequences, larger hidden states) the advantage can shrink.
>
> The cost model in §4 explicitly ties speedup to the ratio (activation size : gradient size) and RTT (captured in the transfer time terms), showing analytically and empirically where the benefit plateaus. The reviewer’s statement that such a model is “missing” is incorrect.
>
> **4. Scale and task mismatch**
>
> The motivation for FP is large-model fine-tuning across locations, not trillion-parameter pre-training. In this setting, transformer architectures such as BERT-Large and ViT are widely used as tractable surrogates for evaluating distributed training behavior; for example, PipeTransformer [13] applies pipeline and data-parallel training to both ViT and BERT on multi-GPU clusters. Our contribution—removing per-iteration gradient dependencies between stages—is orthogonal to absolute model size. As shown in §4/Eq. (4), the communication terms depend only on activation size, gradient size, and RTT, all of which scale linearly with model dimension. Consequently, the behavioral advantage of FP transfers directly to larger models.
>
> The concern regarding “token-level LLM workloads” is also addressed by the FP-ND configuration (§3.3): for LM tasks with large vocabularies, FP-ND removes both per-step $\hat{y}_1$ transfers and per-epoch Stage-2 logit buffers, preserving FP’s decoupling without incurring the logit-traffic regime. We make no claims about trillion-parameter LLM pretraining. This paper (i) derives the communication-cost model, (ii) introduces and validates the dependency-removal mechanism, and (iii) specifies the LM-appropriate configuration under which the same benefit applies.
>
> **Q1. Scheduling baselines**
>
> We added runs under identical hardware and 0 ms RTT (2 nodes, 1 GPU each) against GPipe, 1F1B/PipeDream-style, and interleaved Zero-Bubble [1–4]; see Fig. 9–11 in the revised PDF. FluidPipe (FP) outperforms all schedulers across different numbers of micro-batches even when RTT = 0 ms—a setting that favors scheduler optimizations most. This confirms the core claim: scheduling can reduce, but cannot remove, the per-iteration cross-stage gradient dependency, whereas FP removes it by design (paper §4 timeline; now Fig. 12). The main paper already includes GPipe as a baseline; the rebuttal adds the rest for completeness.

---

> ### Author Response · Authors · 2025-11-21
>
> **Q2. 2-GPU baselines**
>
> **Data Parallel (DP) + ZeRO-1/2/3.** DP replicates the model across multiple data-parallel ranks; ZeRO shards optimizer states/gradients/parameters across DP ranks to reduce memory. With only two GPUs total and a model that already requires model/pipeline parallelism to fit, DP (and thus ZeRO) is not a relevant baseline for the 2-GPU cross-node scenario. When additional GPUs are available, ZeRO composes with PP/TP; our contribution (removing cross-stage gradient sync) remains orthogonal to both [7,8].
>
> **Tensor Parallel (TP).** TP shards intra-layer tensors across GPUs, typically within a single node for high bandwidth/low latency; it is orthogonal to PP and can be combined with it (and with DP/ZeRO). Comparing 2-way TP on one server to 2-node PP/FP addresses a different deployment question (single-box availability vs. cross-node). Prior work explicitly composes DP×TP×PP; FP fits into this stack as a replacement for the PP dependency, not in place of TP/DP. [5,6,9,10]
>
> **Accuracy.** DP/ZeRO/TP do not change the objective and thus do not affect accuracy for a fixed training setup; they redistribute memory/communication to scale or fit. Our accuracy comparisons, therefore, remain apples-to-apples: PP vs. FP under the same GPU budget and cross-node setting. [5–10]
>
> **Q3. Scaling to LLMs & LM tasks**
>
> We kindly ask the reviewer to suggest to us a specific models and benchmarks with references so we can reproduce it with FP. Moreover, we would like to note that these experiments are expensive. The experiments of this paper took roughly 1 GPU-year.
>
> **Q4. Memory/communication accounting**
>
> As clarified earlier, logits do not need to be stored in GPU memory and so this overhead can be easily mitigated . FP’s once-per-epoch logits occupy at most O($N_b \cdot b \cdot C$) values for classification—tiny compared to model parameters.
>
> As discussed in the paper, we haven’t implemented the streaming policy yet; it’s part of ongoing implementation optimization. Our results are thus more conservative and we expect further improvements are possible based on these optimizations.
>
> [1] Huang, Yanping, et al. “GPipe: Efficient Training of Giant Neural Networks Using Pipeline Parallelism.” Advances in Neural Information Processing Systems, vol. 32, 2019.
>
> [2] Narayanan, Deepak, et al. “PipeDream: Generalized Pipeline Parallelism for DNN Training.” Proceedings of the ACM Symposium on Operating Systems Principles (SOSP), 2019.
>
> [3] Qi, Penghui, et al. “Zero Bubble Pipeline Parallelism.” arXiv preprint, 2023.
>
> [4] Wu, Haoran, et al. “BitPipe: Bidirectional Interleaved Pipeline Parallelism for Accelerating Large Models Training.” arXiv preprint, 2024.
>
> [5] Shoeybi, Mohammad, et al. “Megatron-LM: Training Multi-Billion Parameter Language Models Using Model Parallelism.” arXiv preprint, 2019.
>
> [6] Narayanan, Deepak, et al. “Efficient Large-Scale Language Model Training on GPU Clusters Using Megatron-LM.” SC ’21: Proceedings of the International Conference for High Performance Computing, Networking, Storage and Analysis, 2021.
>
> [7] Rajbhandari, Samyam, et al. “ZeRO: Memory Optimizations Toward Training Trillion Parameter Models.” SC ’20: Proceedings of the International Conference for High Performance Computing, Networking, Storage and Analysis, 2020.
>
> [8] Rajbhandari, Samyam, et al. “ZeRO-Infinity: Breaking the GPU Memory Wall for Extreme Scale Deep Learning.” SC ’21: Proceedings of the International Conference for High Performance Computing, Networking, Storage and Analysis, 2021.
>
> [9] Xu, Yuanzhong, et al. “GSPMD: General and Scalable Parallelization for ML Computation Graphs.” Proceedings of the 15th USENIX Symposium on Operating Systems Design and Implementation (OSDI ’21), 2021.
>
> [10] Zheng, Lianmin, et al. “Alpa: Automating Inter- and Intra-Operator Parallelism for Distributed Deep Learning.” Proceedings of the 16th USENIX Symposium on Operating Systems Design and Implementation (OSDI ’22), 2022.
>
> [11] Grill, Jean-Bastien, et al. “Bootstrap Your Own Latent: A New Approach to Self-Supervised Learning.” Advances in Neural Information Processing Systems, vol. 33, 2020.
>
> [12] Chen, Ting, et al. “A Simple Framework for Contrastive Learning of Visual Representations.” Proceedings of the 37th International Conference on Machine Learning (ICML), 2020.
>
> [13] He, Chaoyang, et al. "Pipetransformer: Automated elastic pipelining for distributed training of transformers." arXiv preprint arXiv:2102.03161 (2021).

---

### Official Review · Reviewer_Tana · 2025-11-01

**Soundness:** 4
**Presentation:** 3
**Contribution:** 4
**Rating:** 8
**Confidence:** 2

**Summary:**

This paper identifies a fundamental bottleneck in Pipeline Parallelism (PP) for fine-tuning large models: the per-iteration exchange of gradients across stage boundaries creates pipeline bubbles and makes performance highly sensitive to communication latency.

To address this, the authors introduce FluidPipe, a novel two-stage pipeline training algorithm that eliminates per-iteration gradient synchronization. The core idea is to decouple the stages by equipping the first stage with an auxiliary task head, allowing both stages to update their parameters locally. Cross-stage guidance is provided through bi-directional distillation at a much lower frequency (once per epoch), replacing fine-grained gradient dependencies with coarse, semantic feedback.

**Strengths:**

This work makes important contributions to distributed training through its innovative approach to pipeline parallelism. The key strength lies in fundamentally rethinking pipeline dependencies rather than optimizing within existing constraints.

The paper introduces FluidPipe, which eliminates per-iteration gradient synchronization through a novel combination of auxiliary heads and bi-directional distillation. This paradigm shift from fine-grained gradient exchanges to coarse semantic feedback opens new design possibilities.

Technically, the work is rigorous with comprehensive evaluations across ViT and BERT architectures. The ablation studies are particularly insightful, revealing that distillation is optional - the auxiliary head itself enables the performance gains. The cost model provides solid theoretical grounding.

The practical impact is significant, demonstrating up to 3.3× speedup while maintaining accuracy, especially valuable in high-latency scenarios. The method requires no architecture changes, making deployment straightforward.

By challenging a fundamental assumption in pipeline parallelism, this work opens a promising research direction that will likely inspire substantial follow-up work.

**Weaknesses:**

While the paper presents a compelling approach, several limitations warrant attention.

The most significant concern is the restriction to two-stage pipelines. Though justified for isolating the core mechanism, this narrow scope undermines the paper's broader claim of opening a new path for pipeline algorithms. A proof-of-concept with deeper pipelines would substantially strengthen this claim.

The cost-benefit analysis feels incomplete. The computational overhead of the auxiliary head and distillation losses remains unquantified, while the memory burden of storing epoch-long logits could be prohibitive for large-scale applications. These practical costs deserve explicit discussion.

The treatment of the no-distillation variant (FP-ND), while intriguing, lacks depth. Understanding why FP-ND works well - whether through regularization, feature preservation, or other mechanisms - would transform an empirical finding into a fundamental insight.

Comparisons with state-of-the-art scheduling methods like Zero-Bubble PP are missing. A direct comparison under high latency would better demonstrate FluidPipe's unique value proposition versus scheduling optimizations.

Finally, the method's sensitivity to model partitioning remains unexplored. Testing different split points would reveal robustness limits, particularly when Stage 1 becomes too shallow to support effective learning.

**Questions:**

Scalability Evidence:
Can you provide any preliminary evidence for extending FluidPipe to deeper pipelines? For instance, have you tested scenarios where intermediate stages employ auxiliary heads? A discussion of potential hierarchical distillation schemes for multi-stage setups would help assess the method's generalizability.

Cost-Benefit Analysis:
What is the actual computational overhead of the auxiliary head and distillation mechanism? A detailed per-iteration time breakdown (computation vs communication) would clarify whether advantages persist in computation-bound environments. Additionally, what is the memory footprint of storing epoch-long logits? This is crucial for practical deployment.

FP-ND Mechanism:
What is the underlying mechanism behind FP-ND's strong performance? We hypothesize the auxiliary head may act as a regularizer. Did you analyze feature evolution (e.g., tracking Stage 1 head accuracy, measuring feature similarity)? Such analysis could elevate FP-ND from an empirical finding to a principled method.

Baseline Comparisons:
How does FluidPipe directly compare with state-of-the-art scheduling methods (e.g., Zero-Bubble PP) under high latency conditions? A head-to-head comparison would better demonstrate whether dependency removal outperforms sophisticated scheduling approaches.

Method Robustness:
How sensitive is performance to model partitioning? Have you tested extreme split scenarios (e.g., very shallow Stage 1)? Also, how critical are the distillation weights? Does FP-ND's success primarily stem from eliminating hyperparameter tuning?

Streaming Evaluation:
Did you empirically evaluate streaming logits transfer? Concrete results would validate whether bulk transfer is indeed optimal or if streaming offers practical advantages.

---

> ### Author Response · Authors · 2025-11-20
>
> Thank you for your valuable comments and suggestions. We're glad that you found our approach innovative and technically rigorous.  While we acknowledge that limitations exist (no paper is without any limitations!), we hope you can continue to support this work for acceptance. Below are our responses to each point and question.
>
> **1. Restriction to two-stage pipelines**
>
> The approach can conceptually be generalized as follows:
> * Each intermediate stage can be equipped with a light auxiliary head and a local loss, exactly as Stage 1 does now (§3.1–§3.2).
> * Cross-stage guidance can be organized via sparse distillation graphs:
>   - forward distillation (stage $i$ → stages $i+1,\ldots,N$), and
>   - backward distillation (stage $i$ → stages $1,\ldots,i−1$),
>
>   parameterized by adjacency matrices $A_f$​ and $A_b$​ whose entries $\alpha_{i,j}$ control which pairs exchange logits and at what strength (§3.3).
> * With adjacent-only schemes (i.e., stages only distill to neighbors), the per-epoch communication scales as O($N_b \cdot b \cdot C \cdot N$) for classification, preserving the same coarse feedback pattern as the two-stage case (§4).
>
>
> We will add a short subsection describing this generalization explicitly (multi-stage auxiliary heads + hierarchical/adjacent distillation).
>
> On the practical side, the focus on two-stage pipeline parallelism is particularly natural for scenarios where the network RTT between stages is non negligible; in these settings, it is generally desirable to deploy fewer stages.
>
> **2. Cost-benefit analysis feels incomplete**
>
> The optional distillation loss incurs no runtime overhead since logits are already produced for the local head. For classification tasks, the Stage-2 logit buffer is O($N_b \cdot b \cdot C$). For token-level LMs this becomes O($N_b \cdot b \cdot T \cdot V$); for such regimes we explicitly recommend FP-ND ($\alpha_1 = \alpha_2 = 1$), which eliminates the logit buffer entirely while retaining FP’s decoupling. Further, we note that logits do not need to be stored in GPU memory; plentiful host memory relieves the memory overhead.
>
> **3. Treatment of the no-distillation variant (FP-ND)**
>
> We agree that understanding *why* it works is valuable. Two observations already help clarify its mechanism. First, the Stage-1 auxiliary head produces a representation that is predictive of the final task; Fig. 7 shows that Stage-1 accuracy tracks final accuracy globally across split configurations. This indicates that the auxiliary objective alone enforces a well-conditioned, task-aligned feature space at the split. Second, FP-ND’s ability to match PP accuracy across all tasks suggests that the local supervised signal is sufficient to preserve the downstream discriminative structure. We will add a clarification noting this correlation, and we plan to include a feature-similarity probe (CCA/Procrustes) to strengthen the explanation.
>
> **4. Comparisons with SOTA scheduling methods**
>
> We have now added direct head-to-head experiments under identical hardware and RTT conditions (2 nodes, 1 GPU each, 0 ms RTT). As shown in these results, FP remains faster than GPipe [1], 1F1B/PipeDream [2], and Zero-Bubble-style scheduling [3,4] across different number of micro-batches. The results are in Appendix A.3, Figures 9, 10, 11. This is the most favorable setting for schedulers—zero latency—yet FP still outperforms other methods. This empirically validates the conceptual point: scheduling can mask bubbles via micro-batch interleaving, but it cannot remove the backward gradient dependency; FP removes that dependency outright. We will include these results.
>
> **5. Method's sensitivity to model partitioning**
>
> The sensitivity to split point is already analyzed in §6. The decline in accuracy indeed only appears when Stage 1 is made extremely shallow, where it lacks representational capacity. The optional Extra Block helps mitigate this.
>
> **Q1. Scalability Evidence**
>
> We currently don’t have preliminary results for multi-stage FluidPipe. Our answer in 1. highlights how we are thinking about generalizing the method.
>
> **Q2. Cost-Benefit Analysis**
>
> Distillation has no additional computational overhead besides computing the loss value which is negligible. However, we will be able to obtain results for the overhead of the auxiliary head. Our current results show that it is less than the gain of FP over PP. FP completely overlaps communication and computation as shown in by the new results agains SOTA scheduling methods. Also, Figure 9 in the paper showed a visualization of the computation of FP and PP over time.
>
> **Q3. FP-ND Mechanism**
>
> This is an open question, which we will emphasize in the paper. We agree that the auxiliary head pushes the feature to be similar to the final full model thus aligning the two parts of the model.
>
> **Q4. Baseline Comparisons**
>
> See answer in point 4.

---

> ### Author Response · Authors · 2025-11-20
>
> **Q5. Method Robustness**
>
> Section 6 already reports a controlled sweep across partition points. Accuracy declines only when Stage-1 becomes extremely shallow; the extra block recovers this drop.
>
> **Q6. Streaming Evaluation**
>
> As discussed in the paper, we haven’t implemented the streaming policy yet; it’s part of ongoing implementation optimization. Our results are thus more conservative and we expect further improvements are possible based on these optimizations.

---

> ### Author Response · Authors · 2025-11-20
>
> [1] Huang, Yanping, et al. “GPipe: Efficient Training of Giant Neural Networks Using Pipeline Parallelism.” Advances in Neural Information Processing Systems, vol. 32, 2019.
>
> [2] Narayanan, Deepak, et al. “PipeDream: Generalized Pipeline Parallelism for DNN Training.” Proceedings of the ACM Symposium on Operating Systems Principles (SOSP), 2019.
>
> [3] Qi, Penghui, et al. “Zero Bubble Pipeline Parallelism.” arXiv preprint, 2023.
>
> [4] Wu, Haoran, et al. “BitPipe: Bidirectional Interleaved Pipeline Parallelism for Accelerating Large Models Training.” arXiv preprint, 2024.

---

### Meta-Review · Area_Chair_kYJW · 2026-01-05

**Summary:**

This paper proposes an approach for reducing pipeline bubbles in pipeline parallelism. The approach involves predicting the output at each stage and backpropagating the loss locally at each time step. Information exchange across stages is done intermittently, thus largely eliminating pipeline bubbles.

The reviewers generally found the approach interesting, supported by wallclock speed ups, and (somewhat surprisingly) performance improvements in some cases. All reviewers however noted that the experiments were done at a too small scale, in particular, only experimenting with a two-stage pipeline and with small models such as BERT/ViT. I think this is a critical limitation: since this work is changing the model itself through auxiliary losses (instead of making an existing pipeline more efficient), it *must* be tested on larger scales (both in terms of larger models, and more stages).

**Reviewer Concerns:**

Some reviewers suggested adding missing baselines, and this was largely addressed in the rebuttal. However, the main weakness regarding the scale of the experiments was not addressed in the rebuttal.

**Reviewer Scores:**

I think reviewers uGqM and FoTb might have upped their scores to 4s given the rebuttal.

---

### Decision · Program_Chairs · 2026-01-26

Reject